# Polymorphisms in Genes Involved in Inflammation and Periodontitis: A Narrative Review

**DOI:** 10.3390/biom12040552

**Published:** 2022-04-07

**Authors:** Aniela Brodzikowska, Bartłomiej Górski

**Affiliations:** 1Department of Conservative Dentistry, Medical University of Warsaw, 02097 Warsaw, Poland; 2Department of Periodontology and Oral Mucosa Diseases, Medical University of Warsaw, 02097 Warsaw, Poland; bartek_g3@tlen.pl

**Keywords:** polymorphisms, periodontitis

## Abstract

Current evidence pinpoints that the variability in periodontitis traits in humans may be attributable to genetic factors. Different allelic variants can result in alterations in tissue structure, antibody responses and inflammatory mediators. Consequently, genetic variations may act as protective or risk factors for periodontal diseases. A number of features of the inflammatory and immune response that seem to play a role in the development of periodontitis have a clearly established genetic basis. Identifying genes that contribute to the pathogenesis of periodontitis may be utilized for risk assessment in both aggressive and chronic periodontitis. The aim of this narrative review is to summarize the role of polymorphisms in genes involved in inflammation and periodontitis, including cellular receptors, tissue compatibility antigens, antibodies and cytokines.

## 1. Introduction

Periodontitis is a group of inflammatory disorders, primarily initiated by host interactions with oral bacteria, under the influence of environmental factors. There are many risk factors associated with periodontitis, among which, environmental factors (smoking, pathogens, socioeconomic), hist-related systemic factors (diabetes, other inflammatory conditions, stress, obesity), and genetic factors seem to play a vital role [1]. Rates of disease progression may vary from individual to individual. Genetic predisposition is believed to determine the course of periodontitis, especially in aggressive and rapidly progressing forms [2].

Classifications of periodontal diseases have been subject to substantial discussion for many decades. The 1999 International Workshop on Classification of Periodontal Diseases distinguished between aggressive periodontitis and chronic periodontitis [3]. Consequently, chronic periodontitis affected a high proportion of adults, and was characterized by slow progression. Aggressive periodontitis, on the other hand, was defined as a rare inflammatory condition, characterized by rapid and severe destruction of connective tissue attachment and bone, with minimal presence of microbial deposits that affected younger individuals with familial aggregation. However, the 2017 World Workshop on Classification of Periodontal and Peri-Implant Diseases grouped aggressive periodontitis and chronic periodontitis under a single category, periodontitis, for which a new classification framework was updated to stages (I–IV) and grades (A–C) [4,5]. Staging was based on the severity and the extent of disease, while grading depended on the rate of progression, which was linked with risk factors. In spite of the recent modification in classification, we will refer to them as chronic periodontitis or aggressive periodontitis, as the majority of genetic studies were carried out under the old nomenclature.

The source of variation within a species is genetic polymorphism. Polymorphism is defined as the presence of at least two forms of a gene in a population, each of which occurs with a frequency greater than 1%. Polymorphism is the result of changes in DNA sequence that can be detected by molecular biology methods. Genetic polymorphism can be the result of replacing a single nucleotide in a DNA sequence with another one, removing or inserting one or more nucleotides, or inserting repetitive sequences [6]. Changes can occur in the part of the gene that codes the protein (exon), in the non-coding sequence (intron) or in the transcription-regulating sequence (promoter). Mutations within the exon can change the structure of the coded protein. Changes in the promoter can affect gene transcription, which is the first step in protein formation. Mutations within the intron can lead to a change in gene function [6].

Attempts were made to determine a relationship between the occurrence of genetic polymorphisms, gene expression and susceptibility to certain diseases [7,8]. The aim of these studies was to better understand the etiopathogenesis of the disease and to understand polymorphisms that could serve as markers of susceptibility to the disease. Attention was paid to the fact that the distribution of polymorphisms varies significantly in different populations [6]; therefore, results obtained in one population cannot be directly transferred to another one. With regard to both chronic and aggressive periodontitis, research focused on the polymorphisms of protein-coding genes involved in the inflammatory process and in the immune response, including histocompatibility antigens [9,10], IgG class antibodies [11] and their receptors [12,13,14], as well as CD14 molecules [15,16], toll-like receptors [17], vitamin D receptors [18] and other cellular receptors [19,20,21]. Polymorphisms of genes encoding metalloproteinases [22] and other enzymes [23] have also been studied. However, the greatest attention was paid to the polymorphism of genes encoding pro- and anti-inflammatory cytokines [8].

## 2. Methodology

The relevant studies were obtained through sources available in the NCBI database. The search yielded 1276 results (Figure 1). Those were then filtered to the rsIDs in PubMed. Filtered rsIDs were evaluated manually, taking into account the papers in PubMed researching their influence on genetic polymorphisms. A manual search of PubMed database was also performed to identify other potentially important studies. Special attention was paid to the credibility of the results, highlighting longitudinal studies, systematic reviews and meta-analyses.

## 3. Results

### 3.1. Human Leukocyte Antigens (HLA)

The HLA system is the main human histocompatibility system, playing an essential role in the presentation of foreign antigens to T lymphocytes [9]. It includes three classes of molecules. Class 1, encoded by A, B, C, E, F, G genes, occurs on the surfaces of all nucleated cells and is involved in the recognition of foreign antigens. Class 2, encoded by the DP, DQ, DR genes, is found in antigen-presenting cells. Class 3 includes other proteins involved in immunological processes, including complement components. HLA system genes are closely linked to each other on chromosome 6 and are highly polymorphic. Research on the role of the HLA system in periodontitis focused mainly on the aggressive form of the disease [1,8,10,24] (Table 1). When assessing HLA class II polymorphisms in an ethnically mixed population, it was observed that patients with aggressive periodontitis had at least one of the alleles more often than healthy individuals: DRB1*0401, DRB1*0404, DRB1*0405 or DRB1*0408 [25]. On the other hand, studies on Japanese patients showed more frequent occurrence of the DRB1*1401, DRB1*1501, DQB1*0503, DQB1*0602 alleles and less frequent occurrence of the DRB1*0405, DQB1*0401 alleles in early-onset periodontitis patients than in healthy controls [26]. In a meta-analysis, HLA-A9 and -B15 seemed to represent susceptibility factors for aggressive periodontitis, whereas HLA-A2 and -B5 were potential protective factors against periodontitis among Caucasians [27].

### 3.2. Antibodies (Ig)

Antibodies are glycoproteins that specifically bind to foreign antigens, including bacteria and toxins, with their Fab fragment. By coating the bacterial cell in the process of opsonization, antibodies facilitate its recognition and binding by the phagocytic cell. The main antibodies present in the serum and also the most effective opsonins are IgG-class antibodies [28]. These are divided into four sub-classes: IgG1, IgG2, IgG3 and IgG4. The main sub-class of antibodies produced in response to infection of the periodontium with pathogenic microorganisms is IgG2 [29]. Higher levels of these antibodies were observed in the serum of patients with localized aggressive periodontitis compared to patients with generalized disease, suggesting that higher antibody levels may contribute to limiting the extent of the disease [30]. Significant racial and individual differences in the ability to produce IgG2 antibodies were observed [30,31]. Their increased production was found in individuals who have the n+ allele in the gene, coding the antibody-heavy chain (γ 2 locus), also known as the Gm allele [32]. Studies of Taiwanese patients showed less frequent occurrence of the Gm allele in individuals with chronic periodontitis than in healthy ones, while in the American population, there was no association between periodontitis and the Gm allele [31,32].

### 3.3. Receptor for Antibody Fc Fragment (FcR)

Antibody Fc fragment receptors are found on the surface of host cells capable of binding antibodies [33]. These receptors play an important role in the phagocytosis of antibody-coated microorganisms, in antigen presentation and in the release of inflammatory mediators [33]. Three main groups of receptors for the Fc fragment of IgG were identified: FcγRI (CD64), located on mononuclear phagocytes, FcγRII (CD32), located both on mononuclear and multinucleated cells, as well as in soluble form, and FcγRIII (CD16), including FcγRIIIa on monocytes and macrophages, as well as FcγRIIIb on neutrophils [13]. The combination of stimulatory FcγRIIA and inhibitory FcγRIIB genotypes increased susceptibility to systemic lupus erythematosus and periodontitis in the Japanese population [34]. For three subgroups of receptors—FcγRIla, FcγRIIIa and FcγRIIIb—polymorphisms were identified that can be the cause of the expression of receptors, with different degrees of affinity with antibodies. The occurrence of these polymorphisms was evaluated in relation to periodontitis. In the case of FcγRIla, for which R131 and H131 alleles were identified, studies of Japanese patients did not show differences in the occurrence of individual alleles between patients with chronic [12,35] and aggressive periodontitis [36], on the one hand, and the control population on the other hand. In the Taiwanese population, however, the R131 allele was more common in patients with aggressive periodontitis compared to chronic inflammation. In the Caucasian population, the HI31 allele and the H/H131 genotype were more common in patients with aggressive periodontitis [13], while in patients with chronic periodontitis, a higher incidence of this genotype compared to healthy people was observed only in smokers [37].

Two alleles were also identified for the FcγRIIIa receptor: V158 and F158. Studies on Japanese patients with chronic periodontitis showed more frequent occurrence of the V158 allele in patients with advanced disease than in subjects with moderate disease [35], while patients with recurring disease most frequently demonstrated the F158 allele [38]. In Japanese patients with aggressive periodontitis, no difference was observed in the occurrence of individual alleles compared to healthy individuals [36], while in Caucasian patients, the V158 allele was more common in subjects with aggressive periodontitis [13].

The FcγRIIIb receptor, for which two alleles were identified—NA1 and NA2—is the main FcγR receptor on neutrophils [14]. Both in healthy individuals and chronic periodontitis patients, lower phagocytic capacity of cells was observed in persons with the NA2 allele [39]. Japanese patients with aggressive periodontitis had a higher incidence of NA2 allele compared to the healthy group and the group with chronic periodontitis [36]. There was no correlation between the presence of this polymorphism and the occurrence of aggressive periodontitis in Taiwanese patients or in Caucasian patients [13]. In addition, no relationship was observed between the presence of the NA2 allele and the occurrence of chronic periodontitis in Japanese [12,35], Taiwanese or Caucasian patients [13]. On the other hand, more frequent occurrence of the NA2 allele was found in Japanese patients with recurring chronic periodontitis [12], and in the case of smoking patients, with the NA2 allele, faster disease progression was observed than in those who did not have the allele and to non-smoking patients with the NA2 allele [40]. The NA1 allele, in turn, was found much more frequently among elderly subjects resistant to periodontitis [14].

For FcγR IIIa, a meta-analysis revealed a significant association between specific genotypes with increasing chronic periodontitis and peri-implantitis risks in codominant, dominant, and recessive models [38]. For FcγR IIIb, a significant association between specific genotypes with increasing chronic periodontitis and peri-implantitis risks in codominant, dominant, and recessive models was detected. Taking everything into account, the FCGRIIa (rs1801274), FCGRIIIa (rs396991), and FCGRIIIb (rs1050501) polymorphisms were significantly associated with chronic periodontitis and peri-implantitis and may have a role in the pathogenesis of these diseases.

Less frequently than in the case of receptors for IgG-class antibodies, attention was paid to the role of polymorphisms of the gene encoding receptors for IgA-class antibodies—FcαRI (CD89). Secreted IgA constitutes an important part of the barrier that protects mucous membranes against bacterial infection, participating also in the coating and agglutination of microorganisms, preventing their adhesion and in neutralizing bacterial toxins. A polymorphism of the FcαRI coding gene was identified, consisting of a replacement of adenine (A) by guanine (G) in the DNA sequence in locus nt 324 [41]. G/G homozygotes showed increased phagocytosis of P.gingivalis by both peripheral and gingival fluid granulocytes compared to A/A homozygotes, despite similar FcαRI expression [41]. In addition, the nt 324*A allele was more frequent in patients with aggressive periodontitis than in healthy individuals [41].

### 3.4. CD14 Molecules and Toll-like Receptors

The CD14 molecule is a glycoprotein receptor found—among other sites—on the surface of neutrophils, monocytes/macrophages and fibroblasts, which recognizes bacterial LPS bound to specific LPS-binding proteins circulating in the blood [42]. The LPS/LBP/CD14 complex then acts on the target cell via a toll-like receptor (TLR). To date, 10 TLRs have been identified. The TLR4 receptor mediates activation of the host cell by LPS of Gram-negative bacteria, while TLR2 is a receptor for peptidoglycans and lipoteichoic acid of Gram-positive bacteria, as well as for a hitherto unidentified component of the *Porphyromonas gingivalis* cell wall [17].

Regarding periodontitis, two polymorphisms of the CD14 encoding gene were studied, replacement of cytosine (C) with thymine (T) at position −159, described as −159 (C→T), and −1359 (G→T) [43]. For the −159 (C→T) polymorphism, no association with periodontitis in Japanese patients was observed [42]. Caucasians were found to have more frequent occurrence of the −159*C allele in patients with periodontitis than in healthy individuals, either in the entire study population [15] or only in women [16]. For the −1359 (G→T) polymorphism, the −1359*G allele was more frequently observed in patients with advanced disease than moderate disease [42].

The relationship between polymorphisms of the TLR2 and TLR4 coding genes and periodontitis was also studied, but no relationship was found between the studied polymorphisms and the occurrence of the disease [17]. In a recent meta-analysis, significant association was found between periodontitis and TLR-2 rs1898830 polymorphism under the allelic model (A allele vs. G allele: *p* = 0.014, OR = 1.208, 95% CI: 1.039–1.406), recessive model (GG vs. GA + AA: *p* = 0.028, OR = 0.755, 95% CI: 0.588–0.970), and codominant model (GG VS. AA: *p* = 0.014, OR = 0.681, 95% CI: 0.501–0.925) [44]. In subgroup analysis, TLR-2 rs5743708 polymorphism was associated with periodontitis risk in Asians under an allelic model (G allele vs. A allele: *p* = 0.017, OR = 12.064, 95% CI: 1.570–92.688), dominant model (GA + AA vs. GG: *p* = 0.016, OR = 0.08, 95% CI: 0.010–0.620), and codominant model (GA vs. GG: *p* = 0.016, OR = 1.026, 95% CI: 0.821–1.282). The association between TLR4C > G (rs7873784) allele and chronic periodontitis in Asian patients was found and it may be passed on to offspring, in the form of recessiveness [45].

### 3.5. Vitamin D Receptor (VDR)

Vitamin D plays an essential role in maintaining the balance between calcium and phosphate ions and is necessary for normal bone metabolism [18]. The most active form of vitamin D is 1,25-dihydroxyvitamin D3 (1,25(OH)2D3), which stimulates bone matrix protein synthesis and mineralization [18], as well as stimulating monocytes and macrophages, increasing defense capacity against bacterial infections [18]. The biological activity of 1,25(OH)2D3 is the result of its binding to the VDR receptor [46]. The VDR receptor demonstrates gene polymorphism. VDR polymorphisms are characterized by the presence or absence of sites in the DNA sequence, which are recognized by the restriction endonucleases: Apa I, Bms I, Taq I and Fok I [46]. Regarding periodontitis, research focused on the last three polymorphisms mentioned above. In Japanese patients with chronic periodontitis, the TT genotype and T allele were found to be more frequent for the Taq I polymorphism, characterized by the absence of the site recognized by the restriction endonucleases than in healthy individuals, while the Fok I polymorphism was not associated with the disease [46]. On the other hand, Caucasian individuals were found to have a more frequent occurrence of the t allele (presence of the site recognized by Taq I) in chronic and aggressive periodontitis [18]. There was no correlation between the Bsm I polymorphism and chronic periodontitis, while after a combined analysis of Taq I and Bsm I polymorphisms, a more frequent occurrence of the TB haplotype in patients with chronic periodontitis and Tb in healthy subjects was found [46]. In a very recent longitudinal study, subjects were genotyped for six different bone-related polymorphisms: collagen type Iα1 (COL1A1, Sp1, *Ss* alleles), vitamin D receptor (VDR, Fok I, *Ff* alleles,), calcitonin receptor (CALCR, Alu I, *CT* alleles) and estrogen receptor alpha (ESR1, Pvu II and Xba I, *Pp* and *Xx* alleles) [47]. The association was found between tooth loss and COL1A1 and—in men—CALCR. As a result, presented outcomes contributed to the identification of genes involved in tooth loss and, possibly, susceptibility to periodontitis [47].

### 3.6. Other Cell Receptors

Another cellular receptor whose gene polymorphism was assessed for association with periodontitis is the N-formyl peptide receptor (fMLP). This receptor is structurally analogous to bacterial products that stimulate neutrophils to chemotaxis [33]. In patients with localized aggressive periodontitis, changes in the DNA sequence for fMLP were found compared to patients with chronic periodontitis and healthy individuals. These changes were of two types, namely −329 (T → C) and/or −378 (C → G) [19].

Another receptor whose gene polymorphism was evaluated in periodontitis is the receptor for advanced glycation end products (RAGE). As a result of RAGE interaction on endothelial cells with glycation products, resulting from protein and lipid transformation, an increase in vascular permeability and expression of adhesive molecules was observed, whereas as a result of interaction with monocytes—increased production of cytokines and with fibroblasts—reduced collagen production [20]. In patients with chronic periodontitis, the 1704 (G → T) allele was significantly less frequent than in healthy individuals [20].

### 3.7. Matrix Metalloproteinases (MMPs)

Matrix metalloproteinases (MMPs) are responsible for collagen and extracellular matrix (ECM) degradation of the periodontal tissues. MMPs comprise a family of around 25 members, widely divided into six groups, which are involved in various physiological and pathological processes. Of metalloproteinases, polymorphisms of the MMP-1 encoding gene were most frequently assessed in relation to periodontitis. One of these polymorphisms is characterized by the removal or insertion (G) at position −1607. Alleles were referred to as 1G or 2G, respectively. However, a meta-analysis concluded that single nucleotide polymorphisms of MMP-1 (−1607 1G/2G, −519 A/G, and −422 A/T) were not related to periodontitis risk [22]. 

Other investigated polymorphisms of the MMP-1 encoding gene include −519 (A → G) and −422 (A → T), but no association was found between these polymorphisms and periodontal status [48]. Among others, metalloproteinases for which the relationship between gene polymorphisms and the occurrence of periodontitis was evaluated are MMP-2 and MMP-9 [22,49]. It was suggested that MMP-2 T-790G, MMP-9 C-1562T, and TIMP-2 G-418C gene polymorphisms might be associated with periodontitis in the Taiwanese Han population [49]. Subjects with the genotype of MMP-2 −790 TT or T allele of MMP-2-790T/G, as compared to genotypes of GT + GG genotypes or G allele, were less susceptible to chronic periodontitis (OR = 0.50, 95% CI = 0.25–1.00 and OR = 0.52, 95% CI = 0.28–0.96, respectively). The non-alcohol-drinking participants with C allele of MMP-9 C-1562T, as compared to T allele, were less susceptible to aggressive periodontitis (adjusted OR = 0.4; 95% CI = 0.18–0.90). In a recent meta-analysis, *MMP-2*-753C > T and *MMP-9*-1562C > T polymorphisms were not associated with the risk of periodontitis in the overall population. However, *MMP-2*-753C > T and *MMP-9*-1562C > T polymorphisms had an influence on the susceptibility of periodontitis by ethnicity [50]. The *MMP-9*-1562C > T polymorphism showed a significant association with the risk of periodontitis among Caucasians and the chronic periodontitis/aggressive periodontitis subgroup, whereas *MMP-2*-753C > T polymorphism was significantly associated with periodontitis risk only among Asians.

MMP-8 is another metalloproteinase involved in the pathogenesis of periodontitis. Genetic polymorphisms in the complement factor H (CFH) gene and S100A gene region are strongly associated with serum MMP-8 concentration and its release from neutrophils. Salivary concentrations of S100A8, S100A12, MMP-8, terminal complement complex (TCC), and CFG polymorphisms (rs800292 and rs1061170), were strongly associated with the number of periodontal pockets and alveolar bone loss [51]. The authors concluded that any dysregulation of complement may increase the risk of inflammatory disorders, such as periodontitis. MMPs gene polymorphisms related to periodontitis are depicted in Table 2.

### 3.8. Other Enzymes

Other enzymes whose gene polymorphisms were studied for association with periodontitis include the angiotensin-converting enzyme (ACE) and N-acetyltransferase (NAT2). ACE inactivates peptides, resulting from the initiation of the kallikrein–kinin system and coagulation cascade in the course of inflammation, which cause the formation of mediators that stimulate bone resorption [23]. In the Caucasian population with chronic periodontitis, the occurrence of alleles characterized by deletion (absence of a DNA fragment) and insertion (presence of an additional DNA sequence) in the ACE coding gene was assessed. These alleles were referred to as D or I, respectively. However, there was no difference in the incidence of individual alleles between patients with periodontitis and healthy subjects [23]. Meanwhile, assessing the incidence of the composite genotype, simultaneously including genes for both ACE and TNF-β, it was found that in patients with chronic periodontitis, the DD genotype for ACE and—at the same time—the B2B1 genotype for TNF-β occurred much more frequently [23]. A similar gene dependence for TNF-β was observed for the gene encoding the vasoactive peptide endothelin-1 (ET-1). Among patients with chronic periodontitis, there were more individuals who were heterozygous for both of these genes than among healthy people.

NAT2 participates in processes of detoxification for foreign substances, including arylamines from tobacco smoke, by acetylation. Depending on an individual’s ability to metabolize specific substances, patients can be divided into those in whom acetylation is fast and those in whom it is slow [52]. A correlation was found between NAT2 gene polymorphism and acetylation rate. Acetylation is fast in individuals with one or two NAT2*4 alleles (dominant allele), while it is slow in those with two mutated alleles [52]. It was suggested that individuals with slow acetylation may be more susceptible to smoking-related diseases, especially cancer. The incidence of individual NAT2 genotypes in chronic periodontitis was assessed, for which smoking may also be a risk factor. A much more frequent occurrence of genotypes characteristic for slow acetylation was found in patients with more advanced disease [52].

### 3.9. Cytokines

Interleukin 1 (IL-1) is the cytokine that has received particular attention in research into genetic determinants of periodontal disease. Il-1 participates in a number of processes necessary to initiate and sustain an inflammatory response. It increases the production of adhesion molecules, facilitating leukocyte migration, stimulates the production of the other inflammatory mediators and metalloproteinases, activates T and B lymphocytes, stimulates osteoblasts, leading to bone resorption, and stimulates the programmed death of cells producing an extracellular matrix, thus, limiting the regenerative capabilities of tissues [53]. Different IL-1α polymorphisms may have inverse actions in the pathogenesis of periodontitis [54]. IL-1β, on the other hand, is a notable cytokine biomarker in periodontitis risk, in terms of its development and progression. IL1A and IL1B polymorphisms were reported as noteworthy biomarkers for chronic periodontitis susceptibility in the evaluation of eight meta-analyses, by means of a Bayesian approach [55]. Moreover, there was a statistically significant association of peri-implant bone loss with the homozygotic model of IL-1β (−511) (OR: 2.255; IC: 1.040–4.889) [8].

IL-2 is a pro-inflammatory cytokine, produced, inter alia, by lymphocytes isolated from periodontal tissues, collected from patients with chronic periodontitis [56]. A polymorphism of IL-2 encoding gene was identified—in locus IL-2-330, in the gene promoter DNA sequence, (T) is replaced by (G) [54]. The IL-2 −330G allele had a weak relationship with the periodontitis development (OR: 0.96; 95% CI: 0.72–1.20). In contrast, the IL-2 −330T allele had a strong relationship with the periodontitis risk (OR: 0.85; 95% CI: 0.35–1.24) [54]. In a case-control study, the higher frequencies of the IL2 (+166, −330) haplotype were observed in patients with chronic periodontitis, when compared to controls of a mixed population [56]. In a recent meta-analysis, though, a non-significant association between 330 T/G polymorphism in the IL2 gene and chronic periodontitis, in any allelic evaluation, was reported [57].

IL-4 is a cytokine that strongly inhibits macrophage function, including production of PGE2 and cytokines; hence, it was suggested that a local lack of IL-4 may contribute to periodontal disease progression [58]. In the 1L-4 coding gene, −590 (C → T) polymorphism was identified, as well as intron 2 polymorphism, in which the DNA segment is repeated by 70 base pairs [58,59]. The occurrence of these polymorphisms was assessed in patients with chronic and aggressive periodontitis. In Brazilian patients with chronic periodontitis, there was no difference in the incidence of individual alleles or genotypes compared to the healthy group [60]. In patients with aggressive periodontitis, in the group consisting mainly of Caucasian subjects, the presence of both polymorphisms was found in 1/3 of individuals, while in healthy subjects and those with chronic periodontitis, these polymorphisms were not observed at all [58]. In turn, later studies on Caucasian and Japanese patients with aggressive periodontitis showed differences in the occurrence of polymorphisms, depending on race. There was no difference between the sick and the healthy group, neither in the occurrence of individual IL-4-590 alleles nor in the occurrence of individual IL-4 intron genotypes [59]. However, in the Japanese population, a more common combination of −590 (C → T) polymorphism in the IL-4 promoter with intron 2 polymorphism in healthy subjects than in patients with aggressive periodontitis was observed [59]. A positive association was found between the IL-4R Q551R polymorphism and occurrence of chronic periodontitis in a meta-analysis [61]. The IL-4 −33 CT genotype was negatively associated with the occurrence of aggressive periodontitis.

IL-6 is a multifunctional cytokine involved in tissue destruction in the course of inflammatory response. Polymorphisms −597 (G → A), −572 (G → C) and −174 (G → C) [62] were identified in the IL-6 promoter. The frequency of these polymorphisms was compared in patients with chronic periodontitis and in healthy Caucasian subjects from Europe and Brazil. In the European population, the occurrence of the −572 (G → C) polymorphism was significantly less common in patients with periodontitis than in healthy individuals, while the incidence of −597 (G → A) and −174 (G → C) polymorphisms was similar in both groups [62]. In the Brazilian population, on the other hand, the GG genotype for polymorphism in the IL-6-174 locus was much more common in patients with periodontitis than in healthy subjects, and its incidence increased with the severity of the disease [63]. Moreover, the IL6 -174 “C” allele was protective against periodontitis in the Brazilian population in a recent meta-analysis [64]. It was suggested that the IL-6-174*C allele may play a protective role by reducing IL-6 production [63]. IL-6 rs1800796 may serve as one genetic risk factor for periodontitis patients in the Asian population, especially the Chinese population [65]. G/G genotype of IL-6 rs1800796 was associated with an increased risk of chronic periodontitis in a meta-analysis.

IL-10 is an anti-inflammatory cytokine, whose deficiency or impaired function was suggested as a potential factor increasing susceptibility to periodontal disease [66,67]. The IL-10 gene polymorphism at position −597 seemed to be associated with severe generalized chronic periodontitis in a Turkish population [67]. Another study showed that IL-10-1082 SNP was associated with chronic periodontitis and the IL-10-1082G allele increased the susceptibility to chronic periodontitis in Iranians [68]. IL-10-819 and IL-10-592 SNPs were not related to chronic periodontitis susceptibility, but IL-10-819C and IL-10-592C alleles were slightly higher in those patients when compared to healthy subjects. Multiple logistic regressions revealed that that IL-10-592 AA, -819 TT and ATA/ATA genotype conferred a slight increase in the risk for chronic periodontitis in the Chinese population after adjustment for age, gender and periodontal status [69]. Moreover, a higher quantity of subgingival *Aggregatibacter actinomycetemcomitans* and lower serum IL-10 levels were detected in homozygous ATA/ATA carriers. In the Japanese population, however, there was no difference in occurrence of these polymorphisms between patients with chronic and aggressive periodontitis, on one hand, and healthy individuals on the other one [70]. An increased risk of having gingivitis was found in allele -1082*A-positive children (G/A, A/A); 75% versus 25% in allele A-negative children (G/G). Further, allele A-positive children were at increased odds of having gingivitis of 1.8 (95% CI = 1.05–3.06), compared to allele A-negative children, after adjusting for plaque, age, and gender [71]. The current meta-analysis showed that the IL-10 −592C > A polymorphism was statistically associated with periodontitis risk in the overall population [72]. Moreover, the IL-10 −1082A > G, IL-10 −819C > T, and IL-10 −592C > A polymorphisms were associated with periodontitis risk by ethnicity. The subgroup analysis by ethnicity revealed that the IL-10 −1082A > G polymorphism was significantly associated with periodontitis risk in Caucasians, IL-10 −819C > T polymorphism in mixed population, and IL-10 −592C > A polymorphism in both Asians and mixed populations. Consequently, the IL-10 polymorphisms seem to be of high clinical relevance by ethnicity and would be a useful marker to identify patients who are at higher risk for periodontal disease. On the other hand, the IL-10 promoter rs6667202 (C > A) single-nucleotide polymorphism (SNP) was functional in healthy periodontal tissues [73]. The genotypes AA and AC were related to less expression of IL10 and less production of IL-10, when compared to CC. There was no statistical difference between the genotypes in the subjects with periodontitis, as other factors might have been altering the IL-10’s response. The summary of studies on IL polymorphisms is presented in Table 3.

TNF-α is a pro-inflammatory cytokine present in periodontal tissues in greater quantity at sites of disease progression. The incidence of both single nucleotide polymorphisms and microsatellites in patients with periodontitis was assessed. Most studies conducted on Caucasian, Japanese and mixed-race patients with chronic periodontitis did not show a relationship with periodontal status for polymorphisms −238 (G → A), −308 (G → A), −376 (G → A). −857 (C → T), −863 (C → A), −1031 (T → C), as well as +489 (G → A) [74]. However, Japanese patients were found to suffer more frequently from chronic periodontitis if at least one of the three polymorphisms −1031 (T → C), −863 (C→A) or −857 (C → T) was present. The association between tumor necrosis factor-alpha (TNF-α −308G/A, −238G/A, −863C/A, −1031T/C, and −857C/T) polymorphism and either chronic or aggressive periodontitis susceptibility is conflicting. A recent meta-analysis revealed no significant association between TNF-α −308 G/A SNP and aggressive periodontitis and the risk of aggressive periodontitis development. Moreover, there was no significant association between genotype or allele frequency distribution and clinical manifestations (acute vs. chronic) of aggressive periodontitis [75]. Other meta-analysis, however, supported that variant A of the TNF-α (G −308A) polymorphism may contribute to chronic periodontitis and aggressive periodontitis susceptibility, particularly in Asians and Caucasians [76]. In another meta-analysis, the TNF-α-308G/A polymorphism was significantly associated with decreased risks of chronic periodontitis and aggressive periodontitis in Asians [77]. There were no associations between TNF-α −238G/A, −863C/A, −1031T/C, −857C/T polymorphism and susceptibility to aggressive periodontitis. No associations were found between chronic periodontitis susceptibility and TNF-α −238G/A, −857C/T polymorphism. In another meta-analysis, a significant association was detected between TGF-β1 rs1800469 polymorphism and periodontitis risk (OR = 1.21; 95% CI, 1.05–1.39; *p* = 0.008) [78]. In the subgroup analysis by ethnicity, the significant association was only found among Asians, while no significant association was found among Caucasians. In the subgroup analysis by type and periodontitis, the significant association was only found among chronic periodontitis patients.

In patients with aggressive periodontitis, polymorphisms of the TNFa microsatellite were studied. The TNFa microsatellite, for which 14 alleles were identified, is the most polymorphic of the TNF microsatellites [65]. However, no relationship was found between the studied polymorphisms and the occurrence of the disease [66]. The frequency of gene polymorphism for the TNF receptor type 2, +587 (T → G) was assessed in Japanese patients with chronic periodontitis. This polymorphism was found to be more frequent in patients with advanced disease than in healthy individuals, while no difference was observed between healthy subjects and patients with moderate disease [79]. Clinical and radiological tests confirmed that the disease is more advanced in patients with the G allele [79].

TNF-β, also called lymphotoxin α (LTα), has a similar spectrum of action to TNF-α and is coded by a gene located in the same region on chromosome 6 [66]. The incidence of the +252 (G → A) polymorphism of the TNF-β coding gene was assessed in Caucasian patients with chronic periodontitis [23,74]. American studies on a group of 64 subjects showed no relationship between the assessed polymorphism and periodontitis [74]. However, Czech studies conducted on a 2.5-times larger group showed more frequent occurrence of B2B2 and B1B2 genotypes in patients with periodontitis compared to the control group [23].

TGF-β is a family of polypeptide growth factors involved in the inflammatory process and in regulation of the immune response [80]. Elevated TGF-β1 levels were found in patients with periodontitis, both in gingival tissue and in gingival fluid [57]. However, to date, no relationship has been found between the occurrence of chronic periodontitis and TGF-β1 encoding gene polymorphisms: −509 (C → T), −800 (G → A), −988 (C → A), +869 (T → C) and +915 (G → C) [80]. A meta-analysis suggested that the TGF-β1 −509C/T T allele was associated with decreased risk of chronic periodontitis in Asians, while the +915G/C CC genotype might contribute to increased chronic periodontitis risk in Caucasians [81].

## 4. Conclusions

In summary, periodontitis is a disease whose occurrence depends on the presence of both external (pathogenic microorganisms) and internal (host response) factors at a given time. The quality of the immune–inflammatory response depends, to a large extent, on genetic conditions. Some polymorphisms of genes encoding proteins involved in the host response may be risk factors for inflammatory diseases, including periodontitis. Understanding as many genetic factors as possible related to the host’s response can, therefore, allow a more accurate determination of the disease risk in a given patient; hence, the role of genetic polymorphisms in periodontitis is currently the subject of numerous studies.

## Figures and Tables

**Figure 1 biomolecules-12-00552-f001:**
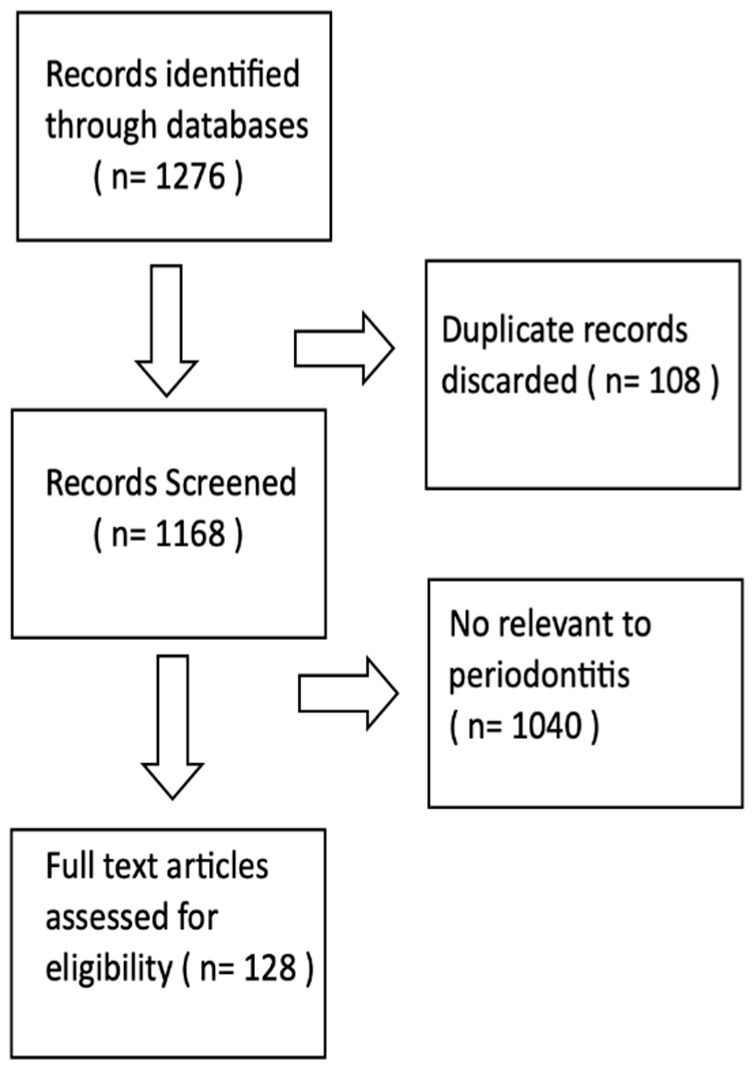
Flow diagram showing article selection process.

**Table 1 biomolecules-12-00552-t001:** Human leukocyte antigens (HLA) gene polymorphisms related to chronic and aggressive periodontitis.

Gene	Position	Periodontitis	Race	Type of Study	References
HLA	DRB1*1501–DQB1*0602	Positively associated with AgP	-	review	[10]
HLA	B*57, DQB1*08, DRB1*04, DRB4*, DQB1*0302	Negatively associated with AgP and CP	German with Caucasian descent	a case control study	[24]
HLA	DR4 (subtypes 0401, 0404, 0405, 0408)	Positively associated with AgP	The ethno-geographic origin of the subjects was neutralized by stratified analysis	a case control study	[25]
HLA	DRB1*1401, DRB1*1501, DQB1*0503, DQBA*0602	Positively associated with AgP	Japanese	a case control study	[26]
HLA	A9, B15 A2, B5	Susceptibility factors for AgP Protective factors against AgP	Caucasian	meta-analysis	[27]

AgP—aggressive periodontitis; CP—chronic periodontitis.

**Table 2 biomolecules-12-00552-t002:** Matrix metalloproteinases (MMPs) gene polymorphisms related to chronic and aggressive periodontitis.

Gene	Position	Periodontitis	Race	Type of Study	References
MMP-1	−1607 1G/2G, −519 A/G, −422 A/T	No associated with the susceptibility to periodontitis	Caucasian, Asian, or mixed (excluding the detailed ethnic results of mixed population in the original study)	meta-analysis	[22]
MMP-2	−1575 G/A, −1306 C/T, −790 T/G, −735 C/T				
MMP-3	−1171 5A/6A				
MMP-8	−381 A/G, +17 C/G				
MMP-9	−1562 C/T, +279 R/Q				
MMP-12	−357 Asn/Ser				
MMP-13	−77 A/G, 11A/12A				
MMP-1	−519 A/G, −422 A/T	No associated with CP	Czech	a case control study	[48]
	−1607 1G	Increased frequency of CP			
MMP-2	−790 T, −790 TT, −790 T/G	Less susceptible to CP	Taiwanese	a case control study	[49]
	−1562 T	Less susceptible to AgP			
MMP-2	−753 C > T	No associated with periodontitis in overall population	Caucasians, Asians, Latinos	meta-analysis	[50]
MMP-9	−1562 C > T				
MMP-2	−753 C > T	Associated with periodontitis in Asians			
MMP-9	−1562 C > T	Associated with periodontitis in Caucasians			

AgP—aggressive periodontitis; CP—chronic periodontitis.

**Table 3 biomolecules-12-00552-t003:** Interleukin (IL) gene polymorphisms related to chronic and aggressive periodontitis.

Gene	Chromosome	Position	Periodontitis	Race	Type of Study	References
Il-1α	2q13-2q21	—889 C/T T	Prevent periodontitis risk	-	meta-analysis	[54]
		—889 C/T C	Strong association with periodontitis development			
		rs1800587	Positively associated with CP	-	meta-analysis	[55]
Il-1β	2q13-2q21	−511 C > T, −3954 C > T	Very trong association with periodontitis development	-	meta-analysis	[54]
		rs1143634	Positively associated with CP	-	meta-analysis	[55]
IL-2	4q26-2	−330 T, −330 G	Positively associated with periodontitis	-	meta-analysis	[54]
		−330T/G	No associated with CP	Caucasian, Asian, Mixed	meta-analysis	[57]
		−166, −330	Positively associated with CP	Mixed	acase-control study	[56]
Il-4	5q31.1	Q551R	Positively associated with CP	German	a case control study	[58]
		promoter, intron, allele 1, allele 2	No associated with AgP	Caucasian, Japanese	a case-control study	[59]
		−590 (C—>T)	No associated with CP	Brazilian	a case-control study	[60]
		Q551R−33 C/T	Positively associated with CP Negatively associated with AgP	Caucasian, Asian, Mixed, Dravidian	meta-analysis	[61]
Il-6	7p21	−572 G/C	Lower susceptibility to CP	Czech	a case-control study	[62]
		−174 G/G	Higher susceptibility to CP	Brazilian	a case-control study	[63]
		−174 G > C	Lower susceptibility to CP	Brazilian	meta-analysis	[64]
Il-10	1q31-32	−824 AA/CC + CA	Positively associated with CP	Turkish	a case control study	[67]
		−1082 G	Positively associated with CP	Iranian		
		−592 AA, −819 TT, −819 ATA/ATA	Positively associated with CP	Chinese	a case-control study	[68]
		−592 C > A	Increased risk of periodontitis	Overall population		
		−1082 A > G	Increased risk of periodontitis	Caucasians	a case-control study	[69]
		−819 C > T	Increased risk of periodontitis	Mixed		
		−592 C > A	Increased risk of periodontitis	Asians, Mixed	meta-analysis	[71]
		−592 C > A	Increased risk of CA	Overall population		

AgP—aggressive periodontitis; CP—chronic periodontitis.

## Data Availability

Not applicable.

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
