# Peer review of "Polymorphisms in Genes Involved in Inflammation and Periodontitis: A Narrative Review"

_biomolecules, 2022, doi:10.3390/biom12040552_

Round 1

Reviewer 1 Report

Dear authors,

The paper should include a summary table with all factors associated   on polymorphisms of protein-coding genes involved in the inflammatory process and in the immune response of the periodontitis.- both chronic and aggressive.

It would be of easier understanding. 

Author Response

Dear Reviewer

Thank you very much for all comments.

The tables have been added to manuscript, as suggested.

I hope that the corrected work will meet the expectations and will be able to be published.

Reviewer 2 Report

The present study proposes a literature review regarding the aspects on the role of genetic polymorphisms in 8 periodontitis.
The work is innovative and presents relevant data on the influence of genetic polymorphisms on the development and maintenance of periodontitis. However, the critical point of the work refers to the methodology used to select the articles included in the review. The methodology was not included, which makes the work extremely fragile and inconsistent.
Thus, in my view, the work, despite presenting an original idea, in the way it is structured, has no possibility of publication.

Author Response

Dear Reviewer

Thank you very much for all comments.

As suggested, methodology has been added. A major revision of the manuscript has been made. I hope that corrected work will meet the expectations and will be able to be published

Reviewer 3 Report

Manuscript: biomolecules-1599208. Genetic Polymorphisms in Periodontitis

The topic is interesting and a review, if well-conducted, helps clarify a debated matter.

However, some major issues need to be addressed before this manuscript can be further considered.

It must be stated that this is a narrative review.

The project is quite ambitious, but not well accomplished. The Author must either implement the paper or select the genes of interest, modifying the title accordingly: polymorphisms in genes involved in inflammation, in bone-related genes (not even mentioned), ….

Review’s definition from guidelines: These provide concise and precise updates on the latest progress made in a given area of research.

Not all the genes discussed show strong association with the disease and that some were investigated in scattered papers, so one can assume that the review covers genes that are only supposedly relevant to periodontitis. Consequently, the present work is not complete in that several genes are not even mentioned. Just to give a few examples: ANRIL, DEFB, TAS2r38, IL-13, MMP-3, GNRH1, PIK3R1, DPP4, FGL2, CALCR, COLIA1, and other bone-related genes.  

Besides its incompleteness, the work is also not updated. Several very good recent papers are missing. The most recent paper cited was published in 2019. This is not acceptable.

Metanalyses are particularly relevant to reviews. In the present work they are referred to on only a couple of occasions: refs 7, 48, and 42 (that does not concern genetics).

Literature search strategy must be described, and possibly reported in a flow chart for clarity.

The Abstract is too short and not informative.

In the Introduction, there is no need to describe what polymorphisms are: lines 17-25 are superfluous.

I’d rather spend a few words in the description of the pathogenesis and different forms of periodontitis.  

Line 30 please replace incidence with distribution.

As the Author states at the beginning of her work, the genetic profile varies across populations, so the context in which the cited studies were conducted must be always reported.

Also, the role of confounding factors as intermediates in the relationship gene-disease -such as smoking, diabetes, etc (when literature is available)- must be discussed.

Each section of the Results must contain:

name of the gene; name and function of the gene product; pathogenetic link with periodontitis; localization, description, and explanation of the evaluated polymorphisms (chromosome, arm, exon/intron, type of polymorphism). The brief introduction so conceived must be followed by the discussion of the available pertinent literature.

I recommend preparing one or two tables in which to summarize: gene name; product molecule; chromosome n, gene position; type of polymorphism; biological effect (if any); association with periodontitis risk/protection/none); list of pertinent references; study population characteristics (size, age, ethnicity). It would be nice to provide the table(s) with SNP ‘rs’ number and OMIM names, in order to move toward a consistent use of the nomenclature across the existing literature.

The whole section concerning IL-1 (and cluster genes, I suppose) is missing, despite being anticipated in line 231.

This work is not accurate with respect to references.

Often the citations do not correspond to the text, either totally or in part. This inaccuracy occurs several times, just to cite a couple: ref. 45 concerns IL4 not CD14; ref. 39 does not concern polymorphisms that are just mentioned at the end of section 3.

They also contain errors, i.e. # 6 and 28 are duplicate, the first using names instead of surnames. The use of first or middle names instead of surnames occurs in more than one occasion.

Moreover, journals’ abbreviations, spaces, punctuation are not consistent throughout the references.

Titles and years of publication contain mistakes.

If a reference occurs more than once in a section, there is no need to cite it several times consecutively (i.e. ref. 26 occurs in lines 75 and 77; ref. 35 in lines 120, 122, 124; ref. 13 in lines 175, 179, 181. Just to mention a few cases. This is not necessary, since there is no other reference in between).

Some reference numbers appear out of the blue, as in the case of #51 that follows #39 (line 198 and 193 respectively).

Ref. 60 (line 270) does not exist. Reference list ends at 56.

References must be available for all international readers. Non-English references must be avoided.

Many pivotal and/or recent papers are missing, in some cases replaced with others either less recent or of a lower level.

Yoshie H, Kobayashi T, Tai H, Galicia JC. The role of genetic polymorphisms in periodontitis. Periodontol 2000. 2007;43:102-32. doi: 10.1111/j.1600-0757.2006.00164.x. PMID: 17214838.

Shaddox LM, Morford LA, Nibali L. Periodontal health and disease: The contribution of genetics. Periodontol 2000. 2021 Feb;85(1):161-181. doi: 10.1111/prd.12357. Epub 2020 Nov 23. PMID: 33226705.

Jakovljevic A, Nikolic N, Jacimovic J, Miletic M, Andric M, Milasin J, Aminoshariae A, Azarpazhooh A. Tumor Necrosis Factor Alpha -308 G/A Single-Nucleotide Polymorphism and Apical Periodontitis: An Updated Systematic Review and Meta-analysis. J Endod. 2021 Jul;47(7):1061-1069. doi: 10.1016/j.joen.2021.03.007. Epub 2021 Mar 26. PMID: 33775731.

Ślebioda Z, Woźniak T, Dorocka-Bobkowska B, Woźniewicz M, Kowalska A. Beta-defensin 1 gene polymorphisms in the pathologies of the oral cavity-Data from meta-analysis: Association only with rs1047031 not with rs1800972, rs1799946, and rs11362. J Oral Pathol Med. 2021 Jan;50(1):22-31. doi: 10.1111/jop.13136. Epub 2020 Dec 11. PMID: 33231892.

Öztürk A, Ada AO. The roles of ANRIL polymorphisms in periodontitis: a systematic review and meta-analysis. Clin Oral Investig. 2022 Feb;26(2):1121-1135. doi: 10.1007/s00784-021-04257-0. Epub 2021 Nov 25. PMID: 34821979.

Author Response

Dear Reviewer

Thank you very much for all comments.

A major revision has been made. The  title and abstract has been changed. Information in the introduction was supplemented in accordance with the comments. Methodology has been added. As suggested , tables have been added to the manuscript. According to the suggestion literature has been supplemented and corrected. Section on  IL-1 have been supplemented. Other errors in the text have been corrected.

I hope that the corrected work will meet the expectations and will be able to be published

Round 2

Reviewer 2 Report

In my point of view, the authors adapted the manuscript according to the suggestions made. However, I suggest making minor review in the English language/grammar.
Thus, after reviewing the English, I consider the manuscript accepted.

Author Response

Dear Reviewer

Thank you for comments.

According to suggestion minor review in the English language and grammar has been made.

I hope corrected work will meet the expectations.